# Co-Carbonized Waste Polythene/Sugarcane Bagasse Nanocomposite for Aqueous Environmental Remediation Applications

**DOI:** 10.3390/nano13071193

**Published:** 2023-03-27

**Authors:** Moonis Ali Khan, Ayoub Abdullah Alqadami, Saikh Mohammad Wabaidur, Byong-Hun Jeon

**Affiliations:** 1Chemistry Department, College of Science, King Saud University, Riyadh 11451, Saudi Arabia; 2Department of Earth Resources and Environmental Engineering, Hanyang University, Seoul 04763, Republic of Korea

**Keywords:** water decolorization, adsorption modeling, waste management, adsorption mechanism

## Abstract

The conversion of worthless municipal solid wastes to valuables is a major step towards environmental conservation and sustainability. This work successfully proposed a technique to utilize the two most commonly available municipal solid wastes viz polythene (PE) and sugarcane bagasse (SB) for water decolorization application. An SBPE composite material was developed and co-pyrolyzed under an inert atmosphere to develop the activated SBPEAC composite. Both SBPE and SBPEAC composites were characterized to analyze their morphological characteristics, specific surface area, chemical functional groups, and elemental composition. The adsorption efficacies of the composites were comparatively tested in the removal of malachite green (MG) from water. The SBPEAC composite had a specific surface area of 284.5 m^2^/g and a pore size of ~1.33 nm. Batch-scale experiments revealed that the SBPEAC composite performed better toward MG adsorption compared to the SBPE composite. The maximum MG uptakes at 318 K on SBPEAC and SBPE were 926.6 and 375.6 mg/g, respectively. The adsorption of MG on both composites was endothermic. The isotherm and kinetic modeling data for MG adsorption on SBPEAC was fitted to pseudo-second-order kinetic and Langmuir isotherm models, while Elovich kinetic and D-R isotherm models were better fitted for MG adsorption on SBPE. Mechanistically, the MG adsorption on both SBPE and SBPEAC composites involved electrostatic interaction, H-bonding, and π-π/n-π interactions.

## 1. Introduction

Synthetic dyes are widely used as coloring agents in many industries. During their manufacturing operations, these industries annually discharge about 10–15% of their synthetic dyes into the aquatic environment [1,2]. Most of the synthetic dyes were toxic, could cause teratogenic and carcinogenic mutations [3], posing potential threats to public health and the ecosystem. Malachite green (MG), a triphenylmethane dye, was used in textiles as a dyestuff and in aquaculture as an antimicrobial agent. MG is known to be harmful to living creatures because of its potential carcinogenicity, mutagenicity, and teratogenicity in mammals and its destruction of the respiratory system, liver, kidney, intestine, gill, and gonads. Because of its noxious effects on living beings, the use of MG in aquaculture applications has been banned in many developed countries. However, due to its efficiency and cost-effectiveness, it is still being commercialized in developing countries [4,5,6]. Thus, for environmental and human health conservation, it is essential to remove or minimize MG concentration in discharged effluents.

To date, several treatment technologies, such as photocatalytic degradation, solvent extraction, membrane filtration, coagulation, and advanced oxidation, have been developed to minimize and/or remove dyes from wastewater [5,6,7,8,9]. Nevertheless, these technologies are often time-consuming, expensive, and, in some cases, ineffective [7]. Adsorption, because of its simplicity, operation ease, and cost-effectiveness, has received much attention in sequestering pollutants from water. Synthesized materials with outstanding physical and chemical properties have been utilized as adsorbents [9,10,11]. Among them, activated carbon, with a high specific surface area, porosity, and loads of surface functionalities, appears to be the most desirable material for the removal of pollutants from water. However, production and regeneration costs limit its wide-scale usage, especially in developing counties.

The activated carbon-based adsorbents developed from agricultural and industrial wastes effectively compete with commercial activated carbon in eliminating organic and inorganic pollutants [8]. Various activated carbon-based adsorbents have been developed by utilizing agricultural and industrial wastes such as *Eucalyptus globulus* seeds [9], maize cob [10], de-oiled soya [11], and forestry waste mixture [12] for the removal of MG dye from the water. Date stone-based activated carbon was developed by Hijab et al. [13] via microwave and thermal treatment methods for the removal of MG dye with an uptake capacity of 98 mg/g from aqueous solutions. 

Sugarcane bagasse (SB) is one of the largest solid industrial waste residues, which can invariably pose a serious environmental concern if not attended to. Annually, more than 279 million tons of SB waste have been generated as by-products, posing potential waste management and disposal issues [14]. According to Rocha et al. [15], the average composition of cellulose, lignin, and hemicellulose in 60 bagasse samples was 42.19, 21.56, and 27.6%, respectively. Chemical analysis revealed that carboxyl, hydroxyl, and phenolic [16] groups were abundantly present over the SB surface. These groups aid in binding heavy metals and dyes [17,18]. Among other wastes, plastic solid waste generation is steadily increasing with time [18,19]. Globally, the random disposal of plastic wastes can lead to freshwater, soil, and ocean contamination [19,20]. Among plastic wastes, polyethylene (PE) is the most dominant universal plastic waste with low monetary value [21]. PE constitutes more than 40% of municipal waste plastic [22]. It is a non-biodegradable waste and, thus, poses a potential threat to ecology and human health. Therefore, it is better to opt for technologies that can convert plastic waste into valuable products. Lian et al. [23] developed graphene/mesoporous carbon electrode materials from the upcycled waste of PE plastic with the addition of GO through low-temperature carbonization at 700 °C. Chen et al. [24] developed activated carbon from polyurethane plastic waste by carbonization and activation for the removal of MG from an aqueous solution with a maximum uptake capacity of 1428 mg/g. Magnetic CoFe_2_O_4_/Co_3_Fe_7_@carbon nanostructures using waste plastics (PET) as the carbon precursor for dye adsorption were developed by Wei and Kamali [25]. The observed adsorption capacities of methylene blue (MB) and methyl orange dyes were 278 and 238 mg/g, respectively. Therefore, the utilization of solid municipal wastes such as plastic and biomass for their conversion to value-added products for energy and environmental applications is one of the best ways toward their management. 

Thus, the aim of the current study was to develop the SBPE composite using waste SB and PE and co-carbonize it at 600 °C under an inert atmosphere to transform it into an SBPEAC composite. Co-carbonization was expected to improve the physicochemical properties of the composite while enhancing its MG removal capacity from aqueous solutions. To the best of our knowledge, the production of activated carbon by blending waste PE and SB and its application in removing MG from an aqueous solution has yet to be reported. Both SBPE and SBPEAC composites were characterized, and their efficacy as an adsorbent was comparatively tested in the removal of MG dye from water. Experimental parameters such as pH, contact time, initial dye concentration, and temperature were optimized through batch-scale study. Additionally, the adsorption data were modeled through isotherm, kinetic, and thermodynamic studies.

## 2. Experimental

### 2.1. Chemicals and Reagents

Malachite green (MG, 99%) and crystal violet (CV, ≥90%) were purchased from Sigma-Aldrich, St. Louis, MO, USA. Methylene blue (MB, >96%) were procured from CDH, Delhi, India. Nitric acid (HNO_3_, 69–71%) and sodium hydroxide (NaOH, 97.5%) were purchased from Merck, Darmstadt, Germany. Hydrochloric acid (HCl, 37%) was obtained from Sigma-Aldrich, Munich, Germany. Deionized (D.I.) water was used for the experiments, collected from the Milli-Q water purification system (Millipore, Burlington, MA, USA).

### 2.2. Characterization 

The BET surface area of the SBPEAC composite was measured using 3Flex (Micromeritics, Norcross, GA, USA). The surface morphologies of SBPE and SBPEAC composites were analyzed by transmission electron microscopy (TEM, Leo 912A 8B OMEGA EF-TEM, Carl Zeiss, Oberkochen, Germany, at 120 keV). The crystallinity of SBPE and SBPEAC composites were analyzed via an X-ray diffractometer (XRD, SmartLab, Rigaku, Austin, TX, USA). The elemental composition of pristine and MG-saturated SBPE and SBPEAC composites was analyzed by X-ray photoelectron spectroscopy (XPS, theta probe-based system, Thermo Fisher Scientific, Waltham, MA, USA). The chemical functional groups present over the surface of pristine and MG-saturated SBPE and SBPEAC composites was analyzed by Fourier transform infrared spectroscopy (FT-IR, Nicolet 6700, Thermo Scientific, Waltham, MA, USA). 

### 2.3. Development of SBPE and SBPEAC Composites 

Sugarcane bagasse (SB) and waste PE bags were procured from a local juice shop and grocery store in Riyadh, Saudi Arabia. The SB was washed with D.I water to remove dirt and traces of sugarcane juice. Thereafter, the SB was manually cut into small pieces. The small pieces of SB were crushed to power in a domestic grinder and sieved to a uniform particle size. Likewise, the PE bags were cut into small pieces, and 0.2 g of PE bag pieces were dissolved in 100 mL toluene at 120 °C through refluxing. SB particles (0.8 g) were suspended into dissolved PE and stirred for 2 h at 100 rpm to uniformly coat PE suspension over SB particles. Thereafter, the air was purged into the suspension to evaporate toluene. Finally, a composite sample with an SB:PE (0.8:0.2 *w*/*w*) ratio nomenclature as SBPE was formed. In addition, the composite sample was co-carbonized through pyrolysis in a silica crucible covered with a lid. During pyrolysis, a sample, nomenclature, such as SBPEAC, was heated for 2 h at 600 °C under an inert atmosphere (nitrogen flow 100 mL/min) with a heating rate of 5 °C/min. An SBPEAC sample yield was 99%.

### 2.4. Adsorption Studies

To determine the optimum conditions for the adsorption of MG, MB, and CV dyes on SBPE and SBPEAC, the influences of various experimental parameters, including contact time (t: 2–600 min), solution pH (2.2–9.5), and initial dye concentrations (C_o_: 20–100 mg/L) were studied through batch scale experiments. The general experimental procedure was as follows: 0.005 g each of SBPE and SBPEAC were separately dispersed in 50 mL of MG, MB, and CV solutions (C_o_: 50 mg/L). The desired pH of the solutions was adjusted using 0.1 M HCl/NaOH solutions. Thereafter, the flasks were equilibrated over a rotatory shaker at 100 rpm for a specified contact time. At equilibrium, the samples were centrifugally separated, and the residual MG, MB, and CV dye concentrations were quantitatively measured using a UV/Vis spectrophotometer (Thermo Scientific, Evolution 600, Madison WI USA) at maximum wavelengths (λ_max_) of 616, 665, and 590 nm, respectively. The respective adsorption capacities (q_e_ mg/g) and the removal efficiencies (R_e_%) of dyes were determined as:(1)$$q_{e},mg/g=\left(C_{o}-C_{e}\right)\times \frac{V}{m}$$
(2)$$R_{e},\%=\frac{C_{o}-C_{e}}{C_{o}}\times 100$$
where q_e_ (mg/g) is the equilibrium adsorption capacity of adsorbents toward dyes, C_o_ and C_e_ (mg/L) refer to the initial and final (equilibrium) dye concentrations, respectively, m (g) is the number of adsorbents, and V (L) is the volume of dye solutions.

## 3. Results and Discussion

### 3.1. Characterization of Composites

The FT-IR spectrum of the SBPE composite displayed a broad peak at 3339 cm^−1^, which was attributed to stretching vibrations of the hydroxyl (-OH) group in different components of SB, such as lignin, hemicellulose and cellulose (Figure 1a) [26]. In addition, this broad peak can also refer to the N-H stretching of aromatic amine materials. After the co-carbonization of the SBPE composite to SBPEAC, a reduction in oxygen and hydrogen atoms containing functionalities was observed. This might be due to the occurrence of the dehydration and decarboxylation reaction at a higher temperature [27,28]. Therefore, the intensity peak associated with the -OH group was significantly reduced, affirming the decomposition of a large number of -OH groups during the pyrolysis [29]. The two strong peaks at 2917 and 2847 cm^−1^ were assigned to asymmetric and symmetric stretching vibrations of the CH_2_ group in PE, respectively [29,30]. The intensity of these two peaks was reduced on the SBPEAC composite spectrum. A peak at 1729 cm^−1^ was due to C=O stretching vibration for the carboxyl and acetyl groups in lignin and hemicellulose. This peak was significantly reduced and shifted to 1697 cm^−1^ in the SBPEAC composite spectrum. The two peaks at 1586 and 1521 cm^−1^ were assigned to C=C stretching aromatic rings present in lignin and cellulose [31,32,33]. These peaks were reduced and shifted to 1563 cm^−1^ in the SBPEAC composite. Peaks at 1460 and 1367 cm^−1^ were attributed to the bending deformation of CH_2_ and methylene deformation in PE [34]. The peaks at 1307 and 1247 cm^−1^ were attributed to the C-N stretching of the aromatic amine [35] and the stretching vibrations of C-OH due to the aryl group in lignin [36], respectively. A peak at 720 cm^−1^ was due to the rocking deformation of methylene groups of PE [34]. A strong peak at 1033 cm^−1^ described the stretching vibration of C-O-C bonds in hemicellulose, lignin, and cellulose [37]. A peak at 894 cm^−1^ was due to C-H deformation in cellulose [34]. The peaks at 1396, 1266, 1010, and 874 cm^−1^ were attributed to the bending deformation of CH_2_, C-N, C-O-C, and Si-O bonds, respectively [37,38]. After the co-carbonization of SBPE composite samples to SBPEAC. The peaks in MG saturated the SBPE composite spectrum and showed an increase in their respective intensities. This indicates that MG was successfully adsorbed onto the SBPE composite (Figure 1a), while MG saturated the SBPEAC spectrum and showed an increase and/or shift to lower wave numbers (Figure 1b). In addition, some new peaks also appeared after MG adsorption. In detail, the adsorption peaks around 3439 cm^−1^ and 1729 cm^−1^ for the O-H and COOH groups stretching vibration increased, respectively, indicating that electrostatic interactions were involved in the MG adsorption onto the SBPE composite. The peak related to the aromatic ring vibration at around 1586 and 1521 cm^−1^ increased, indicating that the π-π stacking effect was involved in MG adsorption onto the SBPE composite. The peak at 1307 cm^−1^ for the C-N stretching of aromatic amine increased in intensity and shifted to 1315 cm^−1^. The peak associated with the C=C group in the aromatic ring was increased in intensity and shifted from 1563 to 1516 cm^−1^, suggesting that π-π stacking was involved in MG adsorption onto the SBPEAC composite. Additionally, a new peak at 1613 cm^−1^ appeared due to the binding of cationic MG dye ion with C=C present on the SBPEAC composite surface. Peaks associated with the Si-O, C-N, and C-O-C groups were decreased in their intensity. Thus, it was concluded that the binding of cationic MG dye over SBPE and the SBPEAC composite surface was a multi-mechanism phenomenon. 

The crystallinity of SBPE and SBPEAC composites was determined through XRD analysis, as illustrated in Figure 1c. A strong characteristic peak at 2θ = 21.55° referred to the reflection of the plane (110) for PE was observed in the SBPE pattern [39,40]. In addition, this peak also indicated the presence of cellulose in SB. Two diffraction peaks at 24.12° and 29.56° referred to as the reflection of the plane (200) and (020) for PE, respectively, in agreement with the previously reported results [41,42]. A weak peak at 36.31°, referred to the reflection of the plane (004), supports the presence of cellulose in SB. All these peaks support the formation of the SBPE composite. After co-carbonization, the strong peaks at 21.55° and 24.12° due to structural deformities in the SBPE composite become very weak. In addition, sharp peaks at 27.45° and 29.51° in the SBPEAC appeared at 600 °C indicating the presence of crystalline mineral forms of SiO_2_ and calcites in SB, which were generated during high-temperature pyrolysis [38,42,43]. These results agreed with the previously reported observations [43,44]. The presence of crystalline minerals forms of SiO_2_ and calcites in the SBPEAC composite was verified through the XPS analysis (details are provided later in this section). Finally, The XRD peaks confirmed that SBPEAC possesses a heterogeneous surface, which reflects the high performance of the SBPEAC composite toward MG adsorption compared to the SBPE composite.

Figure 1d displayed the N_2_ adsorption–desorption isotherm of the SBPEAC composite. Type I isotherm was observed, suggesting the presence of microspores over the SBPEAC composite surface. The Brunauer–Emmett–Teller (BET) surface area of the SBPEAC composite and pore volume was found to be 284.54 m^2^/g and 0.031701 cm^3^/g, respectively. Figure 1d (inset) presents the pore size distributions, which are determined through the BJH model. It was observed that the SBPEAC composite contains micropores with pore size distributions (~1.33 nm), suggesting the possible adsorption of MG molecule with molecular dimensions of 1.21 nm × 1.19 nm × 0.53 nm [45]. 

The morphology of SBPE and SBPEAC composites was analyzed through TEM, illustrated in Appendix A. It was observed that the SBPE composite does not have any pores, whereas, after the co-carbonized of SBPE through pyrolysis, the SBPEAC composite had a porous structure [46]. Similar results were reported by Hossain et al. [47] for mesoporous activated carbon developed from hemp bast fiber through hydrothermal processing.

The full scan XPS survey spectrum of the SBPE composite revealed the existence of three peaks at 284.72, 399.97, and 532.87 eV, which were attributed to C1s (72.1%), N1s (1.5%), and O1s (26.4%) (Figure 2a and Table 1). After the co-carbonization of SBPE to SBPEAC, the atomic concentration of C increased to 82.3%, while a drop in N and O atomic concentrations was observed in Figure 2b. This loss was attributed to the deterioration of functionalities during the pyrolysis of SBPE. In addition, two new peaks at 102.42 and 347.9 eV attributed to Si2p (1.1%) and Ca2p (0.7%) emerged in the SBPEAC spectrum. After MG adsorption on SBPE and SBPEAC, the atomic concentration of C increased from 72.1 to 74.3% and 82.3 to 85.2%, respectively, which might be due to the accumulation of MG ions over the SBPE and SBPEAC surface. The deconvoluted C1s spectrum of SBPEAC showed peaks at 284.61, 285.1, 286.7, and 289.46 eV attributed to C-C/C=C, C-O/C-N, C=O, and O-C=O, respectively (Figure 3a) [46,47]. The high-resolution deconvoluted O1s spectrum of SBPEAC showed four peaks at 531.12, 532.51, 533.80, and 536.01 eV, which were assigned to C-O, C=O, and O-C=O, respectively (Figure 3b) [48]. The deconvoluted Si 2p spectrum of SBPEAC resulted in three peaks located at 101.7, 102.6, and 103.5 eV, assigned to Si-C, Si-O-C, and O-Si-O, respectively (Figure 3c) [49,50]. After MG adsorption on SBPE and SBPEAC composites, the atomic concentration of N1s increased from 1.49 to 1.77% and 0.39 to 4.98%, respectively (Table 1). In addition, the atomic concentration of O1s was reduced from 26.42 to 23.88% and 14.23 to 9.34%, respectively. These results affirmed the successful binding of MG ions over the SBPE and SBPEAC composite surfaces. 

### 3.2. Adsorption Studies

#### 3.2.1. Selectivity Study

The adsorption of three cationic dyes viz. MG, MG, and CV on SBPE and SBPEAC composites under experimental conditions (at C_o_: 25 mg/L, m: 0.005 g, T: 298 K, and t: 24 h) were tested during the selectivity study. The results displayed in Figure 4a reveal that the maximum uptake capacities of MG, MB, and CV dyes on SBPE and SBPEAC composites were 111.5, 95.6 and 79.9 mg/g, and 208.9, 101.8, and 83.4 mg/g, respectively, owing to the difference in their chemical structure in terms of functional groups and molecular size. Among the dyes, the removal efficiencies of MG on SBPE and SBPEAC were 111.5 and 208.9 mg/g, respectively, which was the highest among the tested dyes. The spatial diameters of CV and MB dyes were larger than that of the MG dye as there are two -N(CH_3_)_2_ functional groups in CV and MB but only one -N(CH_3_)_2_ functional group in MG [51,52]. In addition, the large molecule of the CV dye might hinder its binding to the SBPE and SBPEAC surface [53]. MB contains a positively charged S atom, which can bind to negatively charged carbonyl, carboxyl, and hydroxyl groups present on SBPE and SBPEAC composite surfaces. Thus, MB molecules occupy larger areas on the SBPE and SBPEAC surface, hindering the binding of additional MB molecules and consequently reducing its MB removal efficiency [54]. Therefore, MG dye was selected to further understand the adsorption process on SBPE and SBPEAC composites.

#### 3.2.2. Effect of pH

MG is a triphenylmethane-based dye with a pH-dependent molecular structure. The MG structure changes to a carbinol base, losing color under basic conditions due to a reaction that occurs between MG and OH. Therefore, the influence of solution pH on MG adsorption onto SBPE and SBPEAC composites was studied at various pH values, as illustrated in Figure 4b. It was observed that MG adsorption capacities on both SBPE and SBPEAC increased from 6.88 to 111.46 mg/g and 16.69 to 208.76 mg/g with an increasing solution pH from 2.2 to 7.6, respectively. The MG adsorption capacity on SBPEAC was greater than that on SBPE. This might be due to the activation of SBPE to SBPEAC, leading to an increase in oxygen-containing functional groups onto the surface of SBPEAC, resulting in an increase in MG removal at pH 7.6 [24]. On the contrary, in an acidic medium, the adsorption capacities of both the adsorbents were the lowest due to the protonation of both SBPE and SBPEAC surfaces by H^+^ ions. The protonated adsorbent surfaces hindered the binding of cationic MG through electrostatic repulsion. As the solution pH continued to increase, the functional groups on the adsorbent’s surfaces gained a negative charge, attracting cationic MG through electrostatic attraction forces. These results can be confirmed by a point of zero charges (pH_PZC_) plot (Figure 4c). The pH_PZC_ of SBPE and SBPEAC composites were 6.7 and 7.2, respectively. This indicates that the surface of both SBPE and SBPEAC composites were positively charged when pH < pH_PZC_ due to the reaction of H^+^ ions with surface active functional groups, which led to electrostatic repulsion between the positive charge of both SBPE and SBPEAC surfaces and cationic MG dye. This reduces the maximum adsorption of MG onto both SBPE and SBPEAC composites. When the pH > pH_PZC_, the number of negatively charged sites on the SBPE and SBPEAC composites increases. This increase in the adsorption of the cationic MG dye onto both SBPE and SBPEAC surfaces through electrostatic interaction led to an increase in the adsorption capacities of both SBPE and SBPEAC composites toward the MG dye. Similar results were reported for the adsorption of MG on the copper ferrite/drumstick pod biomass composite [55] and magnetite/coir pith-supported sodium alginate beads by Sarkar et al. [56]. Thus, pH 7.6 was selected for further studies. 

#### 3.2.3. Effect of Contact Time 

The effect of contact time on MG adsorption onto SBPE and SBPEAC composites was studied by varying the contact time between 2 and 600 min under fixed experimental conditions (C_o_: 25 mg/L; pH; 7.6; m: 0.005 g; agitation speed: 100 rpm; and T: 298 K), as depicted in Figure 4d. It was observed that MG adsorption onto SBPE and SBPEAC composites was rapid during the initial stages (between 2 and 360 min). After 360 min, it slowed down, reaching an equilibrium in 420 min. The MG adsorption capacities on SBPE and SBPEAC at the equilibrium were 42.31 and 181.28 mg/g, respectively. The initial (first) adsorption stage indicated fast binding of MG to the external surfaces of both SBPE and SBPEAC due to the larger numbers of vacant active sites accessible for MG molecule adsorption. The second stage of adsorption was owed to the diffusion of MG into the pores of SBPE and SBPEAC. Both SBPE and SBPEAC composite saturates during the third and final stage could be explained by the exhaustion of the free adsorptive sites [57]. The adsorption capacity of SBPEAC was comparatively higher than SBPE owing to the larger specific surface area and porous structure of SBPEAC. In addition, the large number of oxygen-containing functionalities present on the SBPEAC surface played a critical role during MG adsorption. Slower adsorption kinetics were observed for the removal of MG using biochar from the spent fermentation sludge of coir wastes with 12.45 mg/g uptake at C_o_: 100 mg/L [58]. The observed equilibration time was 360 min for MG adsorption on *Bacillus cereus* M^1^_16_ with 66 mg/g as an equilibrium adsorption capacity at C_o_: 50 mg/L and T: 303K [59]. Thus, 420 min was selected as the optimum contact time during the study.

#### 3.2.4. Effect of Initial Concentration and Temperature

The impact of the initial concentration (C_o_) on MG adsorption efficiency onto SBPE and SBPEAC composites was studied for C_o_ ranging from 20 to 100 mg/L at varied temperatures: 298, 308, and 318 K. The adsorption capacity of MG on SBPE and SBPEAC composites for the aforesaid C_o_ range increased from 115.6 to 296.3 mg/g and 199.1 to 872.7 mg/g, respectively (Figure 4e,f), while a respective decrease in the removal efficiency from 57.8 to 29.6% and 99.6 to 87.3% at 318 K was observed. The increase in the adsorption capacity at a higher temperature was attributed to an enhancement in the collision energy between MG and the composite surface. Additionally, an increase in the initial MG concentration provided a driving force, which could accelerate the mass transfer of MG onto the surface of SBPE and SBPEAC composites, leading to its enhanced adsorption capacities. The maximum adsorption efficiency of MG on SBPEAC was 99.6%, which was significantly higher than SBPE (57.8%), reflecting the merits of SBPE co-carbonization on MG adsorption. The observed results were comparatively better for MG adsorption on sulfur-doped biochar derived from tapioca peel waste [60] and similar for MG adsorption on pyrolyzed rice husks [61]. Regarding the influence of temperature on the adsorption process, the experimental data revealed that the adsorption efficiency of MG onto SBPE and SBPEAC increased from 50.82 to 57.79% and from 96.51 to 99.57% with a rise in temperature from 298 to 318 K, respectively, reflecting the endothermic behavior of adsorption. This rise in temperature might lead to a reduction in the viscosity of an MG solution, which results in an increase in MG diffusion rate through the pores of the SBPE and SBPEAC composites [62]. These results are consistent with previous studies, which have reported the maximum removal of MG at a higher temperature using reduced graphene oxide [62] and magnetite/coir pith-supported sodium alginate beads [56].

Thus, the results of the influence parameters on the MG adsorption process onto SBPE and SBPEAC composites can be summarized as follows:The solution pH significantly influences MG adsorption on both SBPE and SBPEAC composites with maximum uptake at pH 7.6.The MG adsorption on both composites increases with time, attaining equilibrium at 420 min. The observed adsorption capacities at the equilibrium on SBPE and SBPEAC were 42.3 and 181.3 mg/g, respectively.The adsorption process was endothermic, with maximum adsorption efficiencies of 57.8 and 99.6% on SBPE and SBPEAC at 318 K, respectively.The results reveal that the SBPEAC composite has an excellent ability to remove MG at 80 and 100 mg/L with a 95.7 and 87.3% removal efficiency, respectively.

### 3.3. Adsorption Modeling

#### 3.3.1. Adsorption Isotherm

The data for MG adsorption onto SBPE and SBPEAC composites were modeled by non-linear Langmuir [63], Freundlich [64], and Dubinin–Radushkevich (D-R) [65] isotherm models. Details are provided in the Appendix A. Appendix A) illustrates the MG adsorption isotherms and Table 2 summarizes the calculated isotherm parameters. Based on regression coefficient (R^2^) values, the adsorption of MG onto SBPE at varied temperatures was better described by the DR model, while MG adsorption on SBPEAC was better described by the Langmuir isotherm model in line with previous a study on MG adsorption on graphene quantum dots-vitamin B9-iron (III)-tannic acid [66]. The results confirmed that the adsorption of MG on SBPEAC was established by the monolayer adsorption process with maximum monolayer adsorption capacity (q_m_)—928.6 mg/g at 318 K. The q_m_ for MG uptake over SBPE was 375.6 mg/g at 318 K. The magnitudes of the separation factor (R_L_) during the study were a range from 0 to 1. This affirms the favorable nature of MG adsorption processes onto both SBPE and SBPEAC composites. The increase in K_L_ and K_F_ constants with a rise in temperature confirmed that the MG adsorption on both adsorbents was endothermic [67]. From the DR model, the mean free energy (E) values for MG adsorption on SBPE and SBPEAC were found in the range 0.113–0.129 kJ/mol and 0.727–1.360 kJ/mol, respectively. These values were <8 kJ/mol, indicating the existence of hydrogen bonding forces during the adsorption process [68].

#### 3.3.2. Adsorption Kinetic

The adsorption kinetic of MG on SBPE and SBPEAC composites was investigated to determine the adsorption mechanism of MG on both adsorbents. The non-linear kinetic models, namely, the pseudo-first-order [69], pseudo-second-order, and Elovich [70] models, were applied to kinetics data, and details are given in the Appendix A. The non-linear kinetic plots for MG adsorption onto SBPE and SBPEAC composites are illustrated in Appendix A), and Table 3 displays the calculated kinetic parameters. The experimental data for MG adsorption on SBPE was fitted to the Elovich model, while MG adsorption on SBPEAC was fitted to the pseudo-second-order kinetic model, supported by higher R^2^ values for Elovich and pseudo-second-order models, respectively (Table 3). 

The magnitude of experimental adsorption capacity (q_e,exp._) for MG adsorption on SBPE was very close to the pseudo-second-order model calculated for the adsorption capacity (q_e2,cal._) value. This indicates that the MG adsorption on the SBPE composite may involve the chemisorption process through shared electrons between the MG and SBPE composite. According to the Elovich model, the high value of α (12.333) compared to *β* (0.17317) indicates that the adsorption of MG onto SBPE was feasible [68]. For MG adsorption on SBPEAC, the R^2^ values for the pseudo-first-order kinetic, pseudo-second-order kinetic, and Elovich models were >0.99. In addition, the q_e,exp._ was nearer to q_e1,cal._ and q_e2,cal._ values. This indicates that the MG adsorption on the SBPEAC adsorbent involved both chemisorption and physisorption processes.

#### 3.3.3. Adsorption Thermodynamics

The thermodynamic parameters such as standard entropy change (Δ*S*°), Gibb’s free energy (Δ*G*°), and standard enthalpy change (Δ*H*°) were assessed for the adsorption of MG on SBPE and SBPEAC composites. These parameters were calculated as:(3)∆G°=−RT lnKc
(4)$$\ln K_{c}=\frac{q_{e}}{C_{e}}$$
(5)$$K_{c}=\frac{∆S{}^{\circ}}{R}-\frac{∆H{}^{\circ}}{RT}$$
where *K_c_* is the equilibrium constant [71], *R* refers to the gas constant (8.314 J/mol-K), and *C_e_* is the equilibrium concentration of MG dye. Appendix A displays the Van’t Hoff plots for the adsorption of MG on SBPE and SBPEAC composites, and the magnitudes of thermodynamic parameters are summarized in Table 4. According to the negative Δ*G*° and positive Δ*H*° values, the adsorption of MG on SBPE and SBPEAC adsorbents was spontaneous and endothermic in nature, respectively. A decrease in the magnitude of Δ*G*° was observed with an increase in temperature from 298 to 318 K. This shows that MG adsorption onto both adsorbents was favorable at higher temperatures (318 K) at different MG concentrations, supported by an increase in the adsorption capacities of both adsorbents as the temperature increased. The positive Δ*S*° values revealed the affinity of SBPE and SBPEAC composites towards MG. In addition, it suggests the increased randomness at the solid–liquid interface during the MG adsorption onto both composites. 

### 3.4. MG Adsorption Mechanism

Figure 5 displays the proposed adsorption mechanism of MG onto the SBPEAC composite. The FT-IR and XPS analyses affirmed that the SBPEAC adsorbent-bearing functional groups such as carboxyl, carbonyl, hydroxyl, and aromatic N groups was due to the presence of phenolic compounds, lignin, and cellulose. According to XPS analysis, after MG adsorption, the atomic concentration of nitrogen (N1s) on SBPE and SBPEAC composites increased from 1.49 to 1.77% and 0.39 to 4.98%, respectively (Table 1), while the atomic concentration of oxygen (O1s) was reduced from 26.42 to 23.88% and 14.23 to 9.34%, respectively, verifying that MG dye ions successfully adhered onto SBPE and the SBPEAC composite surface. In addition, the adsorption of MG dye ions on SBPE and SBPEAC composite surfaces was also confirmed by FT-IR analysis (Figure 1). After MG adsorption onto the SBPE and SBPEAC composites, the intensities of most of the peaks were increased and shifted to lower wavenumbers. In addition, many new peaks appeared. In detail, the absorption peaks around 3439 cm^−1^ and 1729 for the O-H and COOH groups stretching vibration increased, respectively, indicating that electrostatic interactions were involved in the MG adsorption onto the SBPE composite. The peak related to the aromatic ring vibration at around 1586 and 1521 cm^−1^ increased, indicating that the π-π stacking effect was involved in MG adsorption onto the SBPE composite. The peak at 1307 cm^−1^ for the C-N stretching of aromatic amine was increased in intensity and shifted to 1315 cm^−1^. The peak associated with the C=C group in the aromatic ring was increased in intensity and shifted from 1563 to 1516 cm^−1^, suggesting that the π-π stacking was involved in MG adsorption onto the SBPEAC composite. Additionally, a new peak at 1613 cm^−1^ appeared due to the binding of cationic MG dye ion with C=C present on the SBPEAC composite surface. The hydrogen bonding interaction between H atoms present on the SBPEAC surface and the N atom of the MG molecule was another possible mechanism for MG adsorption onto the SBPEAC composite, as illustrated in Figure 5. The n-π interaction between the oxygen atom of carbonyl groups (acts as an electron donor) took place on the surface of SBPEAC and the aromatic rings of MG (acts as electron acceptors). These types of interactions were observed during dye adsorption on pomelo fruit peel [72]. In addition, peaks associated with Si-O, C-N, and C-O-C groups were decreased in their intensity. Thus, it was concluded that the binding of cationic MG over SBPE and SBPEAC composite surfaces occurred by different mechanisms such as electrostatic interaction, hydrogen bonding, and π-π/n-π interactions. 

## 4. Conclusions

Activated (SBPEAC) and non-activated (SBPE) composites were developed from SB and PE wastes. The characterization data revealed that the SBPEAC surface was microporous, with a high specific surface area and a large number of surface functionalities. These functionalities played an active role in MG adsorption on SBPEAC. Thus, among the developed composites, SBPEAC showed better MG adsorption (926.6 mg/g) performance under optimized conditions pH: 7.6, t: 420 min, C_o_: 100 mg/L, T: 318 K. Kinetic and isotherm modeling data for MG adsorption on the SBPEAC composite was fitted to pseudo-second-order kinetic and Langmuir isotherm models, while the data for MG adsorption on SBPE was fitted to Elovich kinetic and DR isotherm models.

The MG adsorption on the SBPEAC composite occurred through electrostatic, π–π and n–π interactions, and H-bonding. Thus, the SBPEAC composite, developed through co-pyrolysis of SB and PE wastes, could be considered a sustainable, economical, and eco-friendly adsorbent to effectively remove MG from water. 

## Figures and Tables

**Figure 1 nanomaterials-13-01193-f001:**
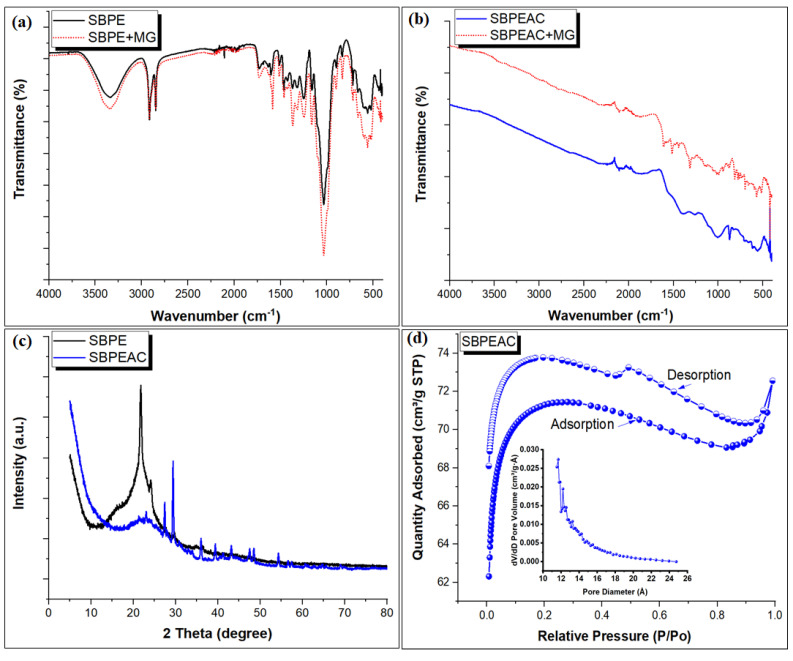
FT-IR spectrum of pristine and MG saturated SBPE (**a**), and SBPEAC (**b**) composites, XRD pattern of composites (**c**), N_2_ adsorption/desorption isotherm of SBPEAC composite, Inset: Pore size distribution curve (**d**).

**Figure 2 nanomaterials-13-01193-f002:**
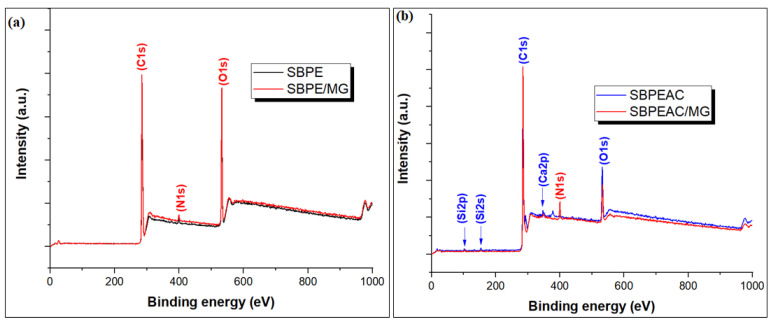
Full scan XPS survey spectra of pristine and MG saturated SBPE (**a**) and SBPEAC composites (**b**).

**Figure 3 nanomaterials-13-01193-f003:**
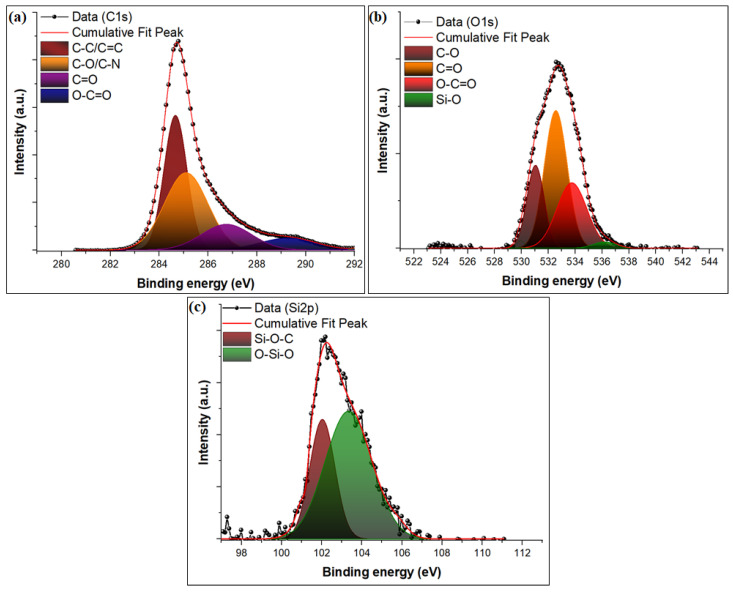
Deconvoluted high-resolution XPS spectra of C1s (**a**), O1s (**b**), and Si2p (**c**) for the SBPEAC composite.

**Figure 4 nanomaterials-13-01193-f004:**
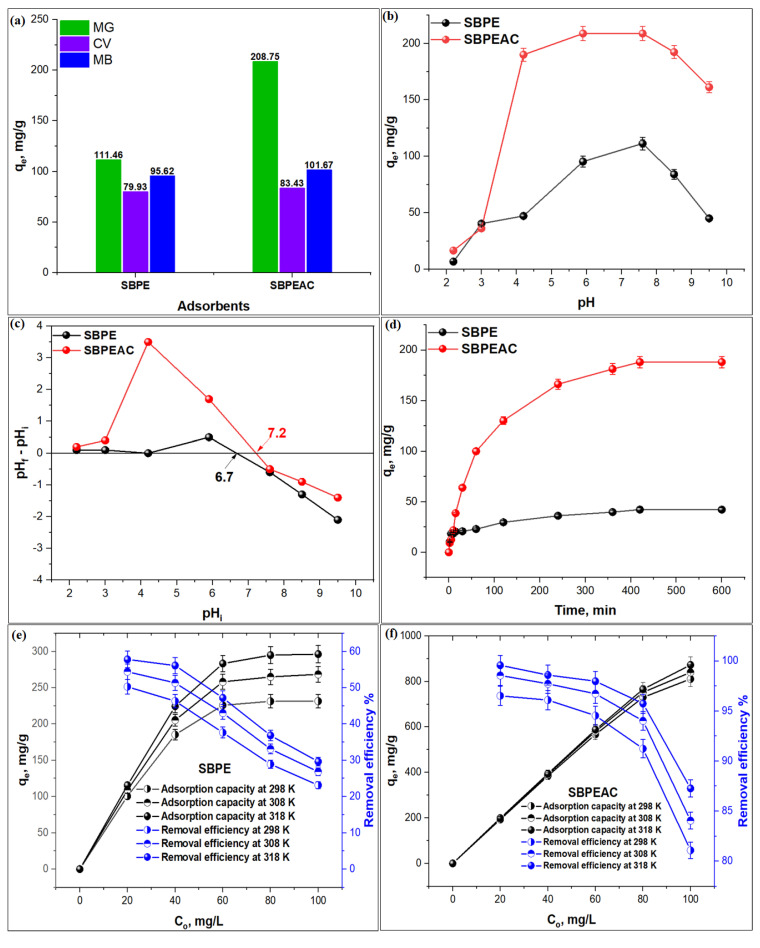
Selectivity study plot of SBPE and SBPEAC composites (**a**), Effect of pH (**b**), Zeta potential (**c**), Contact time (**d**), Initial MG concentration on MG adsorption onto SBPE (**e**), and SBPEAC composites (**f**).

**Figure 5 nanomaterials-13-01193-f005:**
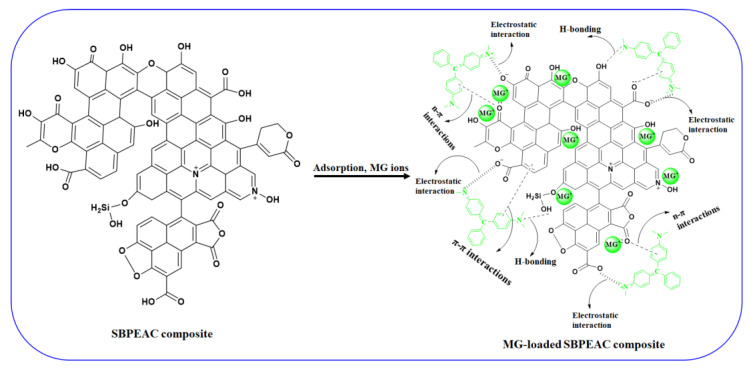
Adsorption mechanism of MG onto SBPEAC composite.

**Table 1 nanomaterials-13-01193-t001:** Element contents of the pristine and MG-loaded adsorbent samples determined by XPS analysis.

Sample	Atomic %						
	C1s	N1s	O1s	S2p	K2p	Ca2p	Si2p
SBPE	72.09	1.49	26.42	0.15	0	0	0
SBPEAC	82.33	0.39	14.23	0.21	1.22	0.73	1.11
SBPE/MG	74.35	1.77	23.88	0.09	-	-	-
SBPEAC/MG	85.24	4.98	9.34	0.07	-	0.44	-

**Table 2 nanomaterials-13-01193-t002:** Non-linear isotherm model parameters for MG adsorption on adsorbent samples at varied temperature.

Sample	Temp. (K)	q_e,exp_. (mg/g)	Isotherm Models										
			Langmuir				Freundlich			DR			
			q_m_ (mg/g)	K_L_ (L/mg)	R_L_	R^2^	K_F_ (mg/g)(L/mg)^1/n^	n	R^2^	q_s_ (mmol/g)	*K_DR_* (mol^2^/kJ^2^)	E (kJ/mol)	R^2^
SBPE	298	231.5	291.1	0.0688	0.420	0.9748	62.07	3.10	0.9343	0.647	39.74	0.113	0.9970
	308	268.4	339.6	0.0681	0.423	0.9741	68.31	2.96	0.9312	0.7442	34.54	0.120	0.9956
	318	296.3	375.6	0.0718	0.411	0.9706	76.66	2.96	0.9249	0.8251	30.15	0.129	0.9969
SBPEAC	298	810.8	905.6	0.4527	0.099	0.9949	340.67	3.11	0.9246	2.063	0.947	0.727	0.9591
	308	840.5	909.7	0.8898	0.053	0.9983	424.24	3.64	0.9272	2.112	0.387	1.137	0.9380
	318	872.7	928.6	1.5920	0.030	0.9791	500.66	4.08	0.9488	2.257	0.270	1.360	0.9072

**Table 3 nanomaterials-13-01193-t003:** Non-linear kinetic model parameters for MG adsorption on adsorbent samples.

Sample	q_e,exp._ (mg/g)	Kinetic Models								
		Pseudo-First Order			Pseudo-Second-Order			Elovich		
		q_e1, cal._ (mg/g)	K_1_ (1/min)	R^2^	q_e2, cal._ (mg/g)	K_2_ (g/mg-min)	R^2^	α (mg/g-min)	β (mg/g)	R^2^
SBPE	42.31	43.17	0.0107	0.5594	41.95	0.00106	0.8347	12.33	0.1732	0.9584
SBPEAC	188.17	185.39	0.0116	0.9933	215.76	0.00006	0.9983	4.45	0.0196	0.9905

**Table 4 nanomaterials-13-01193-t004:** Thermodynamic parameters for MG adsorption on adsorbent samples.

Sample	C_o_ (mg/L)	Δ*H*° (kJ/mol)	Δ*S*° (J/mol-K)	Δ*G*° (kJ/mol)		
				298 K	308 K	318 K
SBPE	20	11.94	59.34	−5.73	−6.35	−6.92
	40	15.49	69.91	−5.34	−6.03	−6.74
	60	15.51	67.03	−4.45	−5.18	−5.79
SBPEAC	20	83.51	326.48	−13.94	−16.70	−20.49
	40	41.29	184.40	−13.64	−15.52	−17.33
	60	40.65	179.29	−12.77	−14.57	−16.36

## Data Availability

Data is available from the corresponding author on reasonable request.

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
