# Peer review of "Co-Carbonized Waste Polythene/Sugarcane Bagasse Nanocomposite for Aqueous Environmental Remediation Applications"

_nanomaterials, 2023, doi:10.3390/nano13071193_

Round 1

Reviewer 1 Report

The manuscript presents a typically ecological topic with considerable application potential. It includes a comprehensive introduction, description of the experimental parts, discussion and discussion of the results, proposed mechanisms, which are illustrated by carefully thought-out drawings. This is one of the few publications that I mu opinion can be published in its current version

Reviewer 2 Report

After having carefully analyzed the above mentioned manuscript, I consider that the study has a significant novelty and the research design is quite interesting. Therefore, I consider that the article must be accept after mayor revision.

1.- The pore texture is not really discussed in the text. For example, the authors should be to provide the volumes of meso and macropores.

2.- As this manuscript relates the mechanism adsorption of the dye on nanocomposites involves electrostatic and π-π/n-π interactions. Nevertheless, important data such as pHPZC (pH at which the surface carbon is neutral, is designated the point of zero charge) are not given in the experimental section or results and discussion. This parameter is important to justify much of evaluation results and discussion (effect of pH as example)

3.- For a better understanding of the adsorption mechanism, the authors should provide the dye species distribution diagram.

4.- The authors should add a discussion paragraph to summarize all the influence parameters and give a more clear conclusion

5.- In figure 2 is missing (b)

6.- The authors should indicate the reason for the presence of S, K and Ca and in the samples.

7.- Line 250 replaced 25 ºC by 298 K

Reviewer 3 Report

The paper focus on the use of Co-carbonized waste polythene/sugarcane bagasse nanocomposite (SBPE) for the removal of malachite green (MG) from water.. The author also investigated the isotherm and kinetic modeling data to pseudo second order kinetic and Langmuir isotherm models. The results mostly support the authors' conclusions. However, some aspects of the manuscript must be carefully reviewed, discussed and improved.

1°) The originality, mechanism, and scientific reliability of the work are unclear. In my opinion, there are some major points that the authors should address before it is accepted for publication.

2°) Why do authors study the removal of this compound: malachite green . Authors must indicate why the removal of this compound could be interesting. 

3°) please give more information about catalyst (MEB, BET, XRD analysis)

4 °) Line 132: The authors indicated that they used UV/Vis spectrophotometer (Thermo Scientific, Evolution 132 600, USA) to measure the concentration of MG. Inf act, the absorption is function of pH . Did they take account the effect of pH on the shifting of absorption values?  

 5°) The part “3.2.4. Effect of initial concentration” should be detailed more to compare the trends with other studies. I suggest adding : Journal of cleaner production 201, 28-38 (2018) ; Sci Rep 13, 1467 (2023).

6) at any moment the authors discussed the results of SBPE regeneration. Thus, what about the reusability of SBPE??

Round 2

Reviewer 2 Report

Accept in present form

Reviewer 3 Report

Authors have addressed all my points. The Ms has improved a lot, very interesting paper actually. I can recommend the Ms for publication now.